# The Possibility of Managed Aquifer Recharge (MAR) for Normal Functioning of the Public Water-Supply of Zagreb, Croatia

Hrvoje Meaški [1] , Ranko Biondić [1,*] , Jelena Loborec [1] and Dijana Oskoruš [2]

1 Faculty of Geotechnical Engineering, University of Zagreb, 42000 Varaždin, Croatia;
  hrvoje.measki@gfv.unizg.hr (H.M.); jelena.loborec@gfv.unizg.hr (J.L.)
2 Croatian Meteorological and Hydrological Service, 10000 Zagreb, Croatia; dijana.oskorus@cirus.dhz.hr
* Correspondence: ranko.biondic@gfv.unizg.hr

**Abstract:** With its quantities of groundwater, the Zagreb aquifer is an irreplaceable water-supply resource that forms the basis of the water-supply of Zagreb, the capital and largest city of the Republic of Croatia. The depth of the Zagreb aquifer system is about 100 m at the deepest part, and the two main aquifers of the aquifer system can be separated vertically by low-permeable clay deposits. In the area of the Zagreb aquifer, there are several active and reserve public water-supply sites, the largest of which are Mala Mlaka and Petruševec. The groundwater level of the Zagreb aquifer is directly related to the water levels of the Sava River, so any erosive change in the Sava riverbed decreases the groundwater levels in the aquifer. In the last 50 years, the groundwater levels in the Zagreb aquifer have decreased significantly, being most pronounced in the area of the Mala Mlaka water-supply site. This has affected the normal functioning of the public water-supply because the suction baskets of the pumps in the dug wells at the Mala Mlaka water-supply site occasionally remain partially or completely in the unsaturated aquifer zone during low groundwater levels, which reduces capacity or prevents pumping from these water-supply facilities. Immediately next to the Mala Mlaka water-supply site is the Sava-Odra Canal, which was built to protect Zagreb from flooding and into which the Sava River flows when its flow rate exceeds 2350 m$^3$/s. This reduces the flow rate of the Sava River near Zagreb and the possibility of flooding urban areas. To prevent problems with groundwater levels at the Mala Mlaka water-supply facilities and to enable normal water-supply, even in extremely dry periods, several variants of managed aquifer recharge (MAR) are proposed here. In order to determine the optimal solution for MAR and to enable the normal functioning of one of the main sites of water-supply in the Zagreb water-supply system. Groundwater flow for the period of 2006 to 2010 was simulated for six different variants of MAR. One assumes a constant potential in the Sava-Odra Canal, three are related to recharge from the Sava-Odra Canal with different backwater levels in the infiltration facility (elevations of 114, 114.5, and 115 m a.s.l.), and two with three absorption wells upstream of the Mala Mlaka water pumping station (injection of 300 L/s each and 500 L/s each). The most favorable method to recharge artificially the Zagreb aquifer near the Mala Mlaka pumping station is achieved with an infiltration facility using an elevation of 115 m a.s.l. The use of such a facility will enable the smooth operation of the water pumping station and the possibility of increasing the pumping quantities at the Mala Mlaka water pumping station for the future development of the area.

**Keywords:** managed artificial recharge (MAR); Zagreb aquifer; alluvial aquifer; mathematical model



## 1. Introduction

The Sava River is the largest tributary of the Danube River. The springing zone is located in Slovenia and has a confluence with the Danube River near Belgrade in Serbia. It flows through Croatia for a large part of its course and forms the border between Croatia and Bosnia and Herzegovina in the eastern part of Croatia. In the wider area of Zagreb,

Croatia, it flows through a deep and irreplaceable aquifer that forms the basis of the water-supply of Zagreb and the neighboring cities and settlements. It is composed of predominantly well-permeable gravels with interlayers of watertight or poorly permeable fine-clastic sediments. This aquifer complex is called the Zagreb aquifer, and it is practically the only real groundwater reserve of Zagreb (Figure 1).

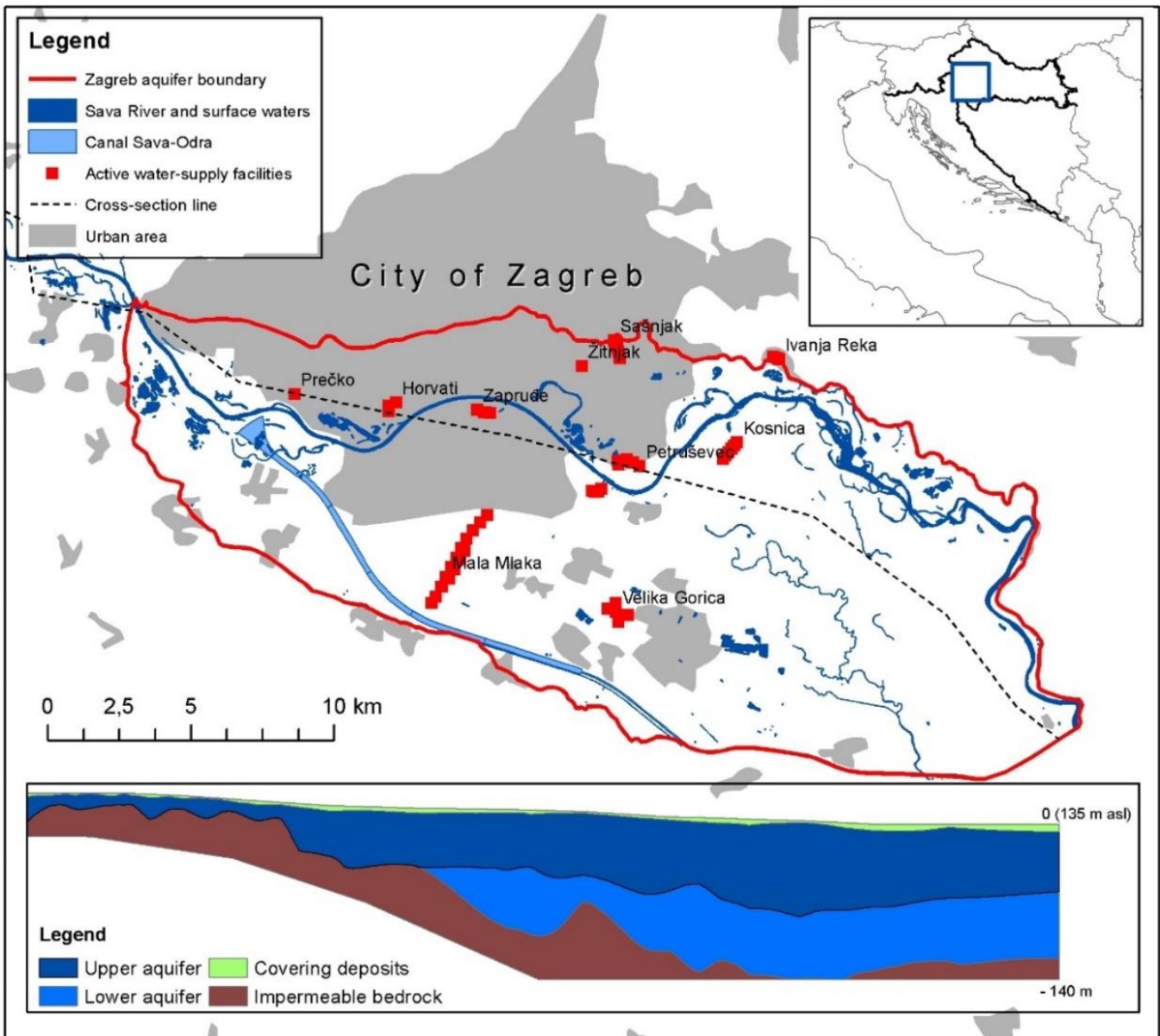

**Figure 1.** Zagreb aquifer position and cross section.

Observing the Zagreb aquifer complex in which permeable and less-permeable layers are present, the vertical component of groundwater flow in some areas is significantly reduced, which is crucial for the protection of the deeper part of the aquifer and determining methods of capturing groundwater. Therefore, the Zagreb aquifer (Figure 1) is in a hydrogeological sense vertically stratified into two aquifers, separated by a low permeable layer of fine-grained clay sediment [1,2]. Detailed hydrogeological characterization is presented later in this paper.

Previous research on the Zagreb aquifer focused on hydrogeological interpretations and determining the connection between the geological structure and the Quaternary aquifer system of the Sava River in Croatia [3], determining the impact of Zagreb landfills

on groundwater quality of the Zagreb aquifer [4], nitrate concentrations in groundwater and the potential impact of agriculture on groundwater quality [5–7], groundwater status and risk assessment for the River Basin Management Plans of the Republic of Croatia [1], interpretations of the content of stable isotopes [8,9], assessment of tritium released from the Krško nuclear power plant (Slovenia) as a groundwater tracer [10], and the reasons for falling groundwater levels in the Zagreb alluvial aquifer [11]. Mathematical modeling of the Zagreb aquifer was performed for the groundwater protection of the Zagreb aquifer [12] and pollution transport [13].

Due to the flood protection of Zagreb in the western part of the study area, the Sava-Odra Canal was built (Figure 1). The Sava-Odra Canal accepts part of the Sava River's high water during rainy periods and prevents flood events that may threaten the urban area of Zagreb. It was built after the great flood of the city of Zagreb in 1964 [14].

Regarding Zagreb's water-supply, it is organized by pumping the Zagreb aquifer with numerous dug and drilled wells. For the water-supply, the upper aquifer is mainly captured using water wells. The water pumping stations in the Zagreb aquifer are: Stara Loza, Prečko, Horvati, Zapruđe, Sašnjak, Žitnjak, Ivanja Reka, Petruševec, and Mala Mlaka; notably, the final works are underway for the Kosnica water pumping. The largest of these are Mala Mlaka and Petruševec (Figure 1).

Previous investigations showed that the groundwater level is directly related to the water levels of the Sava River, erosion of the Sava riverbed, and the extraction of significant amounts of groundwater for public water-supply [1–3,8]. In the last 50 years, the groundwater levels of the Zagreb aquifer have decreased significantly for these reasons. From the 1970s to the present day, the decreases in the western, central, and eastern parts of the Zagreb aquifer have been 1–2 m, 2–5 m, and 1–3 m, respectively. The largest drop level was recorded in the area of the Petruševec and the Mala Mlaka water pumping stations, where this decrease has reached 4–5 m [11].

At the Mala Mlaka water pumping station, the decrease in groundwater levels is affecting the normal functioning of pumping because the bottoms of the suction baskets of the pumps in some wells remain above the groundwater levels during dry periods and during low groundwater levels. In such conditions, these wells cannot be used without pumping water from deeper-drilled wells located in the immediate vicinity of the pumping wells.

In order for the public water-supply to function normally, without the need to include these additional drilled wells, thinking has moved toward managed artificial recharge (MAR), which would locally raise the groundwater level and allow normal water-supply. Several methods of artificial infiltration are used worldwide [15–17]: the infiltration spreading method through controlled flooding [18]; channel modifications by the construction of seepage barriers, absorption wells, canals, and wells [19,20]; and infiltration induced through riverbanks and the use of runoff. In addition to surface water or rainwater, treated wastewater can also be used for managed artificial recharge [21].

Within this paper, we present several variant solutions for the managed artificial recharge of the Zagreb aquifer in the zone of the Mala Mlaka water pumping station. These are artificial recharge by infiltration from the artificial Sava-Odra Canal with a constant water level of 0.5 m, recharge by raising the artificial deceleration at elevations of 114, 114.5, and 115 m a.s.l. as well as the design of three recharge wells with a capacity of 300 or 500 L/s each [22].

The effect of individual artificial feeding was examined through a mathematical model of the Zagreb aquifer developed for this purpose. The model was created in the FEFLOW finite element method software package. For calibration purposes, 53 piezometers were selected that were evenly distributed throughout the Zagreb aquifer, and special attention was paid to the narrow area of the Mala Mlaka water pumping station, where piezometers located next to the pumping wells were selected as calibration points.

## 2. Mathematical Model of the Zagreb Aquifer

The hydrogeological system of the Zagreb aquifer (Figure 1) is a complex natural system, which was simplified using a mathematical model for meaningful analysis. The final aim of the mathematical model was to capture the distribution of potentials (piezometric water levels) and to simulate groundwater flow for the purpose of selecting the best MAR solution.

The input parameters for the mathematical model of the Zagreb aquifer were groundwater levels, water levels of the Sava River, precipitation amounts, and pumping quantities for the needs of the public water-supply. Groundwater levels were determined based on data from 392 measuring points (piezometers) in the study area. The water levels of the Sava River in the investigated area were measured at a total of nine gauging stations uniformly distributed from the entrance profile to the model on the western part of the model to the exit from the model in the eastern part of the model. To construct the mathematical model, we used data from four rain gauging stations that covered the model area well.

Prior to the development of the mathematical model of the groundwater flow in the Zagreb aquifer, a conceptual model of the Zagreb aquifer was defined, based on existing research and databases, i.e., geological and hydrogeological characterization, the geometry of the aquifer system, types of the aquifer, porous medium properties, recharge and discharge places of the system, boundary conditions of the model, and initial conditions in the model.

### 2.1. Geological and Hydrogeological Characterization

The wider investigated area is built of Quaternary age deposits in the lowland area along the Sava River in the vicinity of Zagreb. Quaternary age deposits are sedimented in a morphologically irregular area of separated deep basins. In the western part, out of the investigated area, the Samobor aquifer is located with a maximum depth of about 50 m, followed by the Podsusedski threshold where Quaternary deposits are in some places only 3 m thick. After this threshold follows the Zagreb basin, which gradually deepens and in the deepest part in the area of Črnkovac, it reaches a thickness of about 100 m [1,2], while in the eastern part of Zagreb the aquifers are not as deep. This Zagreb basin is called the Zagreb aquifer. The eastern of the Zagreb aquifer and near the town of Sisak is the new threshold, where the Sisak aquifer is located. The Samobor and Sisak aquifers were not considered in this study, but their citation is important for a comprehensive understanding of the position of the Zagreb aquifer.

The Zagreb aquifer formed during the Middle and Upper Pleistocene when this area was a lake or a swamp area. From the surrounding land area, through the processes of erosion and denudation, the deposits were eroded and washed away by numerous streams into that lake or swamp area. At the beginning of the Holocene, the Sava River breached and then the transport of materials from the Alps began. The water inflow in the hypsometrically lower Zagreb basin led to a drop in water energy and the creation of accumulation conditions with the deposition of a large amount of predominantly well-permeable gravel up to 60 m thick, above the previously deposited finely grained swamp sediments (Figure 2). This contributed to the additional vertical separation into two aquifers having different hydraulic characteristics, separated by impermeable or poorly permeable deposits [23,24].

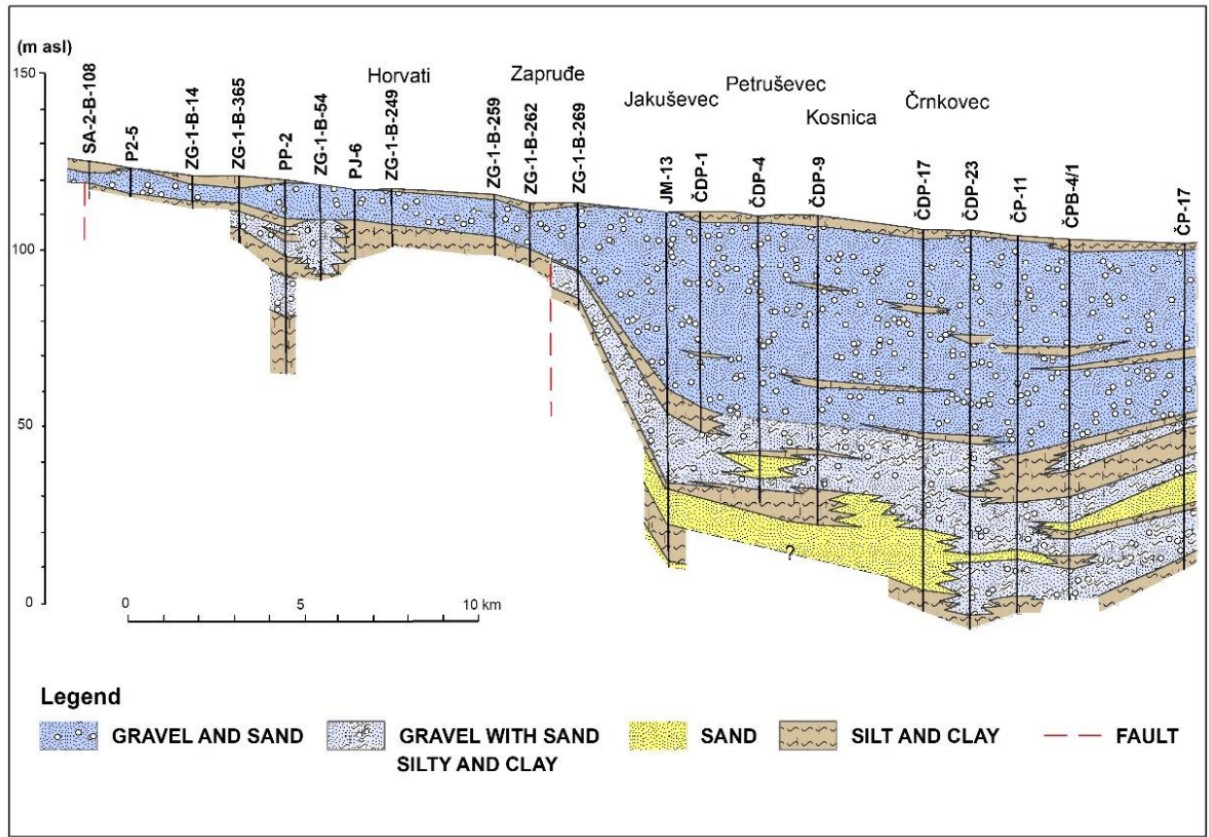

**Figure 2.** Hydrogeological cross section [24].

The above-mentioned aquifer geometry was confirmed with a large number of piezometric wells that were drilled in the Zagreb aquifer system. There are also extensive data available on lithology by aquifer depth. The piezometric wells were geologically determined by piezometric well logs data that describe lithology in intervals.

Overall, the hydrogeology complexes were defined using over 1360 piezometric wells and based on 445 selected characteristic piezometers that well reflect the geometry of the aquifer and its lithological relations. It is significant that many of these wells show the entire profile of aquifers, from covering deposits and gravel aquifers to bottom impermeable deposits (clay and marl). In some places, the wells are not evenly distributed because, e.g., the concentration of wells is higher in the locations of existing and planned water pumping stations.

The initial values of the spatial distribution of the hydraulic conductivities for the aquifer complex were obtained entirely based on discrete values because there were no other data. The existing database on hydraulic conductivity was obtained from Croatian Waters (state agency for water management), and lithological data were obtained from individual piezometers [2] or several hydrogeological studies related to determining the hydrogeological parameters of the Zagreb aquifer [23,25–29]. The database on hydraulic conductivity contains approximately 350 discrete values of hydraulic conductivity, determined as part of water research works in the Zagreb aquifer.

The values of specific yield and porosity for individual aquifers have not been as well researched in the Zagreb aquifer; therefore, the required values were determined based on lithological data from existing piezometric wells and partly obtained from the literature [30–32].

The spatial distribution of hydrogeological parameters over the entire model domain was made by interpolating discrete values collected from verified piezometric wells.

### 2.1.1. Covering Deposits

The surface of the investigated area is covered by a relatively thin humus layer that is 0.2–0.3 m thick, below which is a fine-grained clay layer 0.5–2.0 m thick near the Sava River watercourse; moving away from the river the layer thickness increases to about 10 m and is composed of particles of powder, clay, and fine-grained sand of the Upper Pleistocene to Holocene age. In some places, this poorly permeable layer is missing, and therefore, the aquifer is completely open to the surface.

Those cover deposits are important for the protection of aquifers, but within the investigated area, their spatial distribution does not represent a significant barrier. Hence, it was concluded that the aquifer below them is an open aquifer.

The spatial distribution of hydrogeological parameters of covering deposits was determined so that all cover deposits were assigned the same mean effective value of individual parameters. By averaging the values, the spatial distribution of the parameters of the cover deposits was greatly simplified, which did not significantly affect the groundwater flow simulation itself and had a positively affected the groundwater flow simulation speed. According to available data, the average effective value of hydraulic conductivity of 10 m/day was determined for cover deposits.

### 2.1.2. Upper Aquifer

The upper aquifer has an uneven thickness that ranges from 5 m on the threshold to a maximum of 60 m in the area of the Petruševec and Črnkovec water pumping stations. Downstream from the investigated area its thickness gradually decreases to 10–20 m.

According to available data, the upper aquifer consists of well-compacted unsorted gravel with a sand content of 20–30% and powder up to 5%. The sediment grain size is larger along the Sava River, whereas toward the east, the grain size decreases and gradually turns into sand. Locally, there are layers and lenses of dusty sand 0.2 to 1.3 m thick.

In the western parts of the aquifer and near the Sava River watercourse, the hydraulic conductivity is over 3000 m/day, while in the central parts of the aquifer, it is around 2000 m/day. In the eastern part of the upper aquifer, gravel is completely absent, and only sand is present. Therefore, the modeled hydraulic conductivity decreases to about 300 m/day [33].

### 2.1.3. Aquitard

At a depth of about 30 m, a weakly permeable clay layer with an average thickness of about 2.5 m (1–11 m) is spread over the largest part of the aquifer area. This was determined on numerous piezometric wells. However, it is not present in some places; therefore, it is an incomplete layer.

The hydraulic conductivity of this layer was estimated to be $K = 1 \times 10^{-6}$ m/s. Due to its hydrogeological characteristics, when present, it divides the aquifer system into two parts.

### 2.1.4. Lower Aquifer

Below the weakly permeable clay layer is a lower aquifer about 20 to 30 m thick. The increased part of fine-grained components, reducing the hydraulic values of the lower aquifer and the groundwater, is also under moderate pressure.

The hydraulic conductivity of the lower aquifer layer was estimated based on research conducted in the wider area of Črnkovec and Kosnica water-supply sites. In the central part of the lower aquifer, the value of hydraulic conductivity is about 750 m/day; toward the east, it is significantly lowered to less than 100 m/day. This is about 30% less than that of the upper aquifer.

### *2.2. Conceptual Model*

The boundaries of the model are defined by the geological structure located between the mountainous area in the north (Medvednica Mt.) and the hilly part in the south, which

stretches from Podsused in the west to Rugvica in the east. A similar model domain has been used in other previous studies of the Zagreb aquifer [23,25].

Based on the previously described Zagreb aquifer complex, the conceptual hydro-geological (hydrostratigraphic) model of the Zagreb aquifer system was defined by the existence of two aquifers separated by a thin incomplete aquitard. The extension, thickness, and hydrogeological properties of these layers within the model domain are inconsistent.

Hydrogeologically, the Zagreb aquifer is an open aquifer, for which the upper limit of saturation is also the free groundwater level. In some places, poorly permeable cover deposits are present. However, they do not significantly affect the groundwater flow in a conceptual sense, but it is still possible that minor anomalies may occur in groundwater flow simulations, especially in the parts of the Zagreb aquifer where the impact of cover deposits is somewhat higher. These parts of the Zagreb aquifer can then be considered semiconfined aquifers.

Mathematical simulations of groundwater flow require defining the hydrogeological parameters of a particular aquifer. In accordance with the conceptual model of the Zagreb aquifer system, we processed the available data necessary for the assessment of the hydro-geological properties of the cover deposits, the upper aquifer, and the lower aquifer. The parameters primarily included the assessment of the values of the vertical and horizontal hydraulic conductivity (K) of individual aquifers, specific yield (Sy) for open aquifers, and the effective porosity (n) of individual aquifers in the Zagreb aquifer. For the needs of the model, the hydraulic conductivity Ky is equal to the hydraulic conductivity Kx, whereas the value of the hydraulic conductivity Kz is based on available data and previous research, taken as 10-times smaller values.

The analysis of equipotential and water levels of the Sava River in the predicted period of groundwater flow simulation in the Zagreb aquifer, i.e., in the period from 2006 to 2010, showed that the aquifers are recharged mostly through the infiltration of water from the Sava Riverbed. The groundwater flow direction is generally from west to east or southeast depending on hydrological conditions. The marginal parts of aquifers are hydraulically impermeable in the north, the main inflow is from the west, and the outflow from the aquifer occurs in the eastern part. Research indicated that there are certain inflows of varying intensity along the southern boundary of the model [34], which was considered during the development of the model.

A certain infiltration of precipitation into the aquifer system should be considered, as well as the underground inflow of water from the upstream area and the inflow from the southern hilly area. Comparing groundwater levels from piezometers and data on water levels from the gauging station on the Sava River, a strong correlation was observed, especially for piezometers closer to the Sava River. As the distance from the Sava River increases, this correlation decreases but is still present.

The above is confirmed by previous calculations of the water balance of the Zagreb aquifer, according to which the contribution of the Sava River to the restoration of ground-water was estimated to be about 73% [35].

### 2.3. Boundary Conditions

The boundary conditions of the water flow model were set in accordance with the conceptual model of the Zagreb aquifer system and additionally defined by the spread of less permeable deposits on the edge parts of the Zagreb aquifer.

The four types of boundary conditions of the model (main inflows and outflows from the model domain) and boundary conditions within the model domain (influence of the Sava River, active water pumping stations, and the impact of the Sava-Odra canal) were defined as shown in Figure 3.

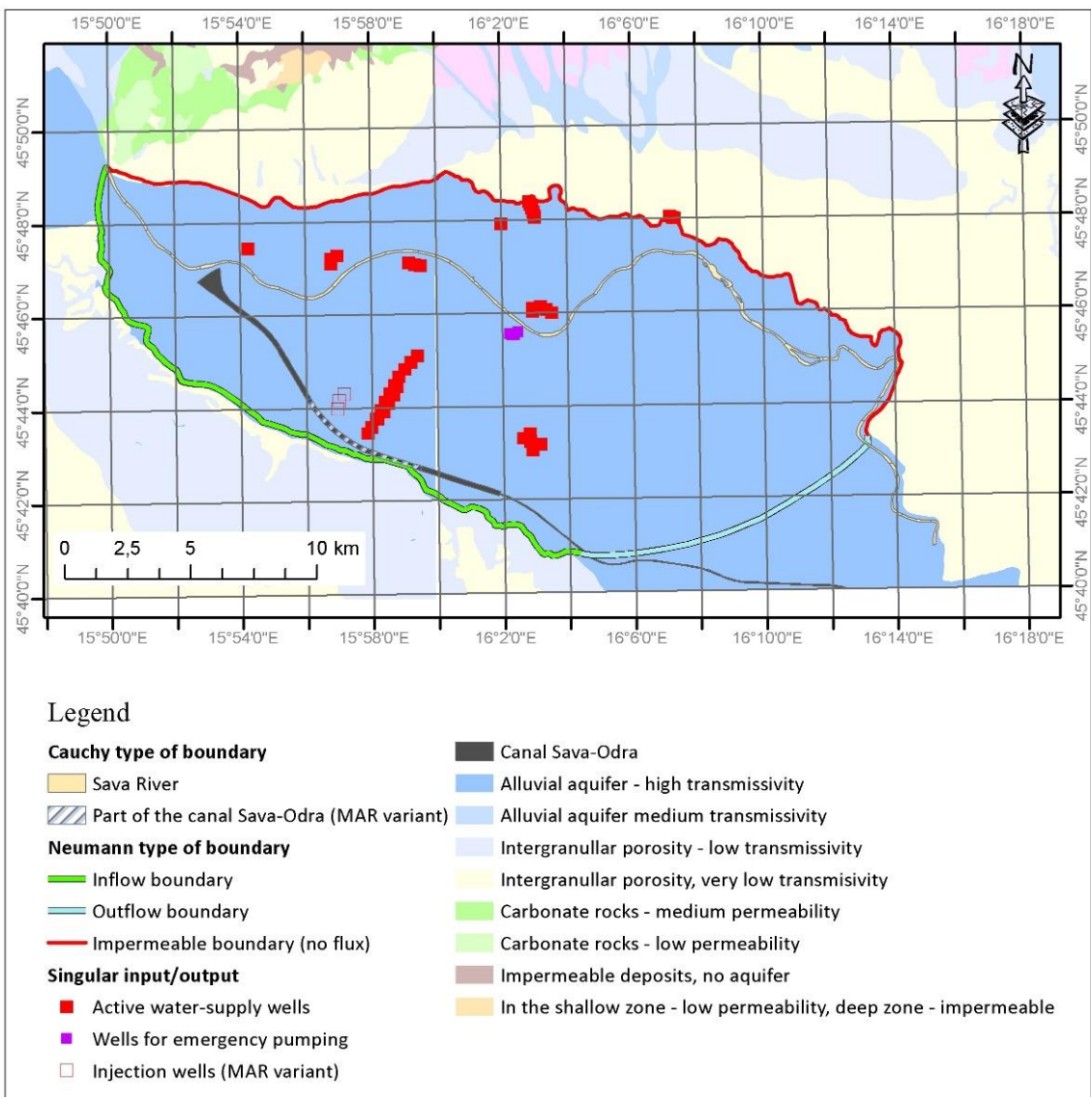

**Figure 3.** Hydrogeological map and boundary conditions defined in the Zagreb aquifer model.

The northern boundary of the model domain (inflows from the direction of Medvednica Mt.) is defined as practically impermeable, although there are numerous, mostly smaller, torrents along it that form in the submountain zone, but are practically all channeled by the infrastructural development of Zagreb. Therefore, they do not have a direct impact on the Zagreb aquifer. If there is any impact on the Zagreb aquifer, then it is mostly smaller and occasional; hence, it would not significantly affect the groundwater flow simulation.

The main groundwater inflow into the Zagreb aquifer is defined in the western part of the model boundary, which is related to inflows from the upstream part of the Samobor aquifer. There are also several smaller inflows along the southern boundary of the model, along which a smaller inflow into the Zagreb aquifer was identified in several places [34]. The main runoff from the system is defined on the southeastern boundary of the model domain, from the Sava–Odra Canal area, across the Odra watercourse, to the Sava River downstream of Zagreb. All boundary conditions of the model domain were simulated by the second type of boundary conditions (Flux- or Neumann-type boundary).

The infiltration of total precipitation in the Zagreb aquifer is defined by the second type of boundary conditions, as a model input over the entire surface of the model domain.

2.3.1. Role of the Sava River within the Boundary Conditions

In the initial simulations of groundwater flow, the Sava River was simulated along its entire length as the first type of boundary condition (head- or Dirichlet-type boundary). Conceptually, this was justified because it is a surface water flow that was assumed to be in strong good hydraulic connection with the Zagreb aquifer; therefore, its effect could be simulated in the form of known piezometric potential. However, shortly after initial groundwater flow simulations, we noticed that its impact is not so simple that it could be simulated only using a Dirichlet-type boundary and that the impact of the Sava River is one of the crucial factors affecting the total groundwater flow.

The Sava River does not represent a watercourse that intersects the entire aquifer system along its entire depth so that the hydraulic connection of the Sava River with the Zagreb aquifer system is not uniform. Therefore, it was ultimately simulated over the entire domain of the Sava River flow model using the third type of boundary condition (transfer- or Cauchy-type boundary). This type of boundary also represents the dependence of the model on the piezometric potential (water level), but it is also conditioned by the resistance to water flow that occurs during the infiltration of water from the Sava riverbed into the aquifer, and vice versa. This means that the Sava River influences the surrounding aquifer (and vice versa), which is realized through the transition medium, the clogging layer, a certain hydraulic conductivity, and thickness.

Therefore, the conceptual model was corrected and revised, and the Sava River was divided into a total of 34 separate segments. The segments represent a uniform distribution of the entire watercourse and were selected based on previously interpolated data from gauging stations on the Sava River. In the processes of successive simulation and calibration of the mathematical model, transfer in/out relations were determined for each segment. In the end, a satisfactory solution of the Sava River's influence on the water flow in the Zagreb aquifer was obtained.

2.3.2. Water Pumping Station Locations

In the area of the Zagreb aquifer, several water pumping stations for public water-supply have been built for the needs of the cities of Zagreb and Velika Gorica. This large aquifer contains large reserves of groundwater, but Zagreb being located directly on the aquifer and the cover deposits being relatively thin (up to several meters) create difficult conditions for protecting the quality of groundwater in the area. Part of the water pumping station is active, part is spare, and part is, unfortunately, abandoned due to the pollution of significant parts of the Zagreb aquifer.

The Mala Mlaka water pumping station is located on the right bank of the Sava River in the central part of the Zagreb aquifer. It is one of the oldest and largest water pumping stations in Zagreb. Although the idea for its construction started back in 1934, construction began in 1956, and the first wells started operation in 1964. In the first phase, 10 dug wells with a diameter of 6 m and a depth of 13 to 15 m were constructed, which provided about 1.7 m$^3$/s of drinking water, which was more than enough for the needs of the city at that time. Subsequently, due to the lowering of groundwater levels, an additional six wells were drilled, which affected deeper parts of the alluvial aquifer and the pumped water is transferred to the dug wells. On average, about 1.3 m$^3$/s is produced from the water pumping station.

Data on pumping quantities used in the mathematical model of the Zagreb aquifer refer to the water pumping stations of Mala Mlaka, Petruševac, Velika Gorica, Sašnjak, Žitnjak, and Zapruđe, which were active during the modeled period.

Finally, the existing water pumping stations within the Zagreb aquifer were simulated using the fourth type of boundary conditions (singular inputs/outputs within the model). Within the mathematical simulation, this type of boundary condition was separately defined for each well within each water pumping station, considering the position of the well, pumping quantity, the permeability of the surrounding deposits, and the exact position of the filter within the well (multilayered-well definition).

### 2.4. Model Discretization

2.4.1. Spatial Discretization

The horizontal and vertical discretization of the model domain is conditioned by the conceptual model of the Zagreb aquifer and the boundaries of the model. It was constructed made using a FEFLOW finite element network generator, which creates super-elements, i.e., geometric frames of finite elements. In the case of the Zagreb aquifer, three-node triangular elements for 2D model representation and six-node prisms with a triangular base for 3D model representation were selected for discretization. The number of these super elements was determined based on factors that could influence the behavior of the mathematical model in later modeling, such as water pumping station locations, the total size of the model domain and individual super elements, conceptualization of natural conditions, and computer processor speed. Discretization achieves the required heterogeneity, i.e., greater or lesser discretization of individual parts of the model domain. In the case of the Zagreb aquifer, spatial discretization of the model was increased near the Sava River, boundaries of the model, larger lakes, existing active water wells in the Zagreb aquifer, as well as in the wider area of Mala Mlaka and the Sava–Odra Canal (Figure 4). In the conceptual sense, the Zagreb aquifer is three-dimensionally divided into three separate layers: cover deposits, and upper and lower aquifer.

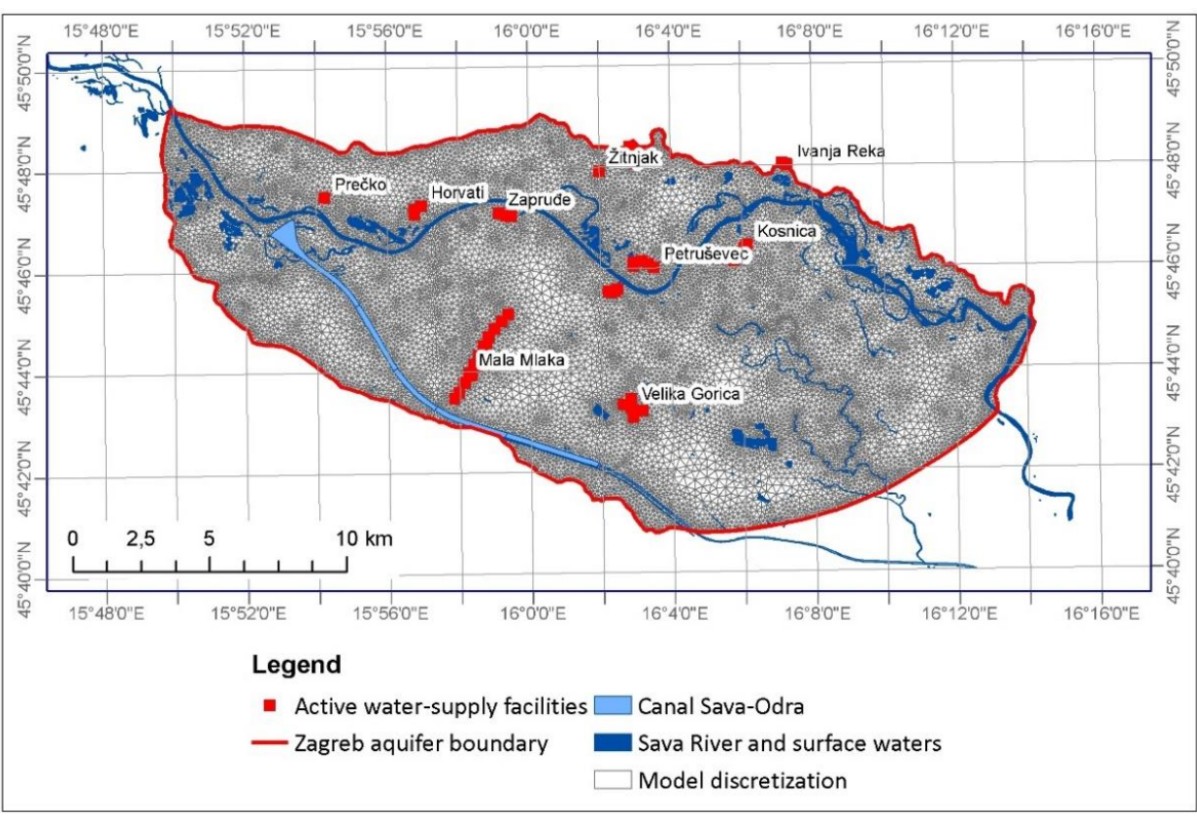

**Figure 4.** Discretization of the Zagreb aquifer model.

2.4.2. Temporal Discretization

The temporal discretization of the mathematical model was performed based on the time observation frequencies of three key factors influencing the simulation of groundwater flow in the Zagreb aquifer: Sava River water level, groundwater level at the piezometers, and frequency of pumping volume measured at water pumping stations. The availability, frequency, and uniformity of these measurements in the Zagreb aquifer are not the same everywhere.

The water level of the Sava River is one of the parameters we used in the simulation of the groundwater flow of the Zagreb aquifer. In addition, the available time series of inflows and outflows at the domain boundaries were also considered. The estimates for the inflow and outflow in the mathematical model of groundwater flow are based on the measured groundwater levels near the individual model boundary, i.e., the recorded time series of these measurements.

The groundwater level measuring frequency at the piezometers in the Zagreb aquifer is not uniform. In some piezometers daily measurements are available, in others, there are only two to three measurements per week. Available measurements of pumping quantities at active water pumping stations of the Zagreb aquifer are defined on a monthly basis. Therefore, within the model, these data posed the biggest problem in the temporal discretization of the model. Initially, the time interval was set to one day, but a comparative analysis of the simulation results for a time interval of one day and five days showed no differences. Therefore, a time interval of five days was selected for the final time interval.

*2.5. Initial Conditions*

To solve the nonstationary problem of groundwater flow, it was necessary to define the initial conditions in the model, i.e., the initial distribution of potential (groundwater level) in the Zagreb aquifer for a certain date. The required levels were obtained by searching a database containing time series with measured groundwater levels; in the case of the Zagreb aquifer, these included a piezometric well database and a database of gauging stations on the Sava River.

Measurements of groundwater levels in the Zagreb aquifer began in early 1972 and continue to this day. The series of groundwater levels on several piezometers in the area of the Mala Mlaka pumping station were analyzed. In the period from 1972 to 2019, two characteristic periods with different trends were observed. The first occurred from the beginning of 1972 to 1994, when there was a continuous decrease in groundwater levels in the Zagreb aquifer; after 1994, the groundwater level stabilized, and no further decline was recorded. Continuous decrease in the first period was the result of several different factors. In the initial part of that period, an embankment was built along the Sava River for flood protection, which prevented the flooding of the coastal area but also reduced the infiltration into the aquifer. The second reason could be attributed to the construction of reservoirs and hydropower facilities upstream from Zagreb in neighboring Slovenia. The third reason could be the increasing the amount of pumping rate for public water-supply in Zagreb in that period. All those reasons led to the constant lowering of groundwater levels and some industrial objects, such as thermal power plants, started to have problems with using the water for cooling purposes. Due to that, an artificial river threshold was built in 1993, which caused the stabilizing of the groundwater levels. Looking at the period from 1993 until today, there were no new regulations of the Sava River in the area of Zagreb, and pumping quantities were not significantly increased.

For the period of the mathematical model, a period of five years was chosen (2006–2010) in which there were no large water level variations, i.e., when groundwater levels were low but steady (Figure 5).

After necessary corrections of the input values and using the Kriging interpolation method, a map of groundwater levels was obtained for the initial time of the simulation of the water flow model in the Zagreb aquifer for 1 January 2006 (Figure 6).

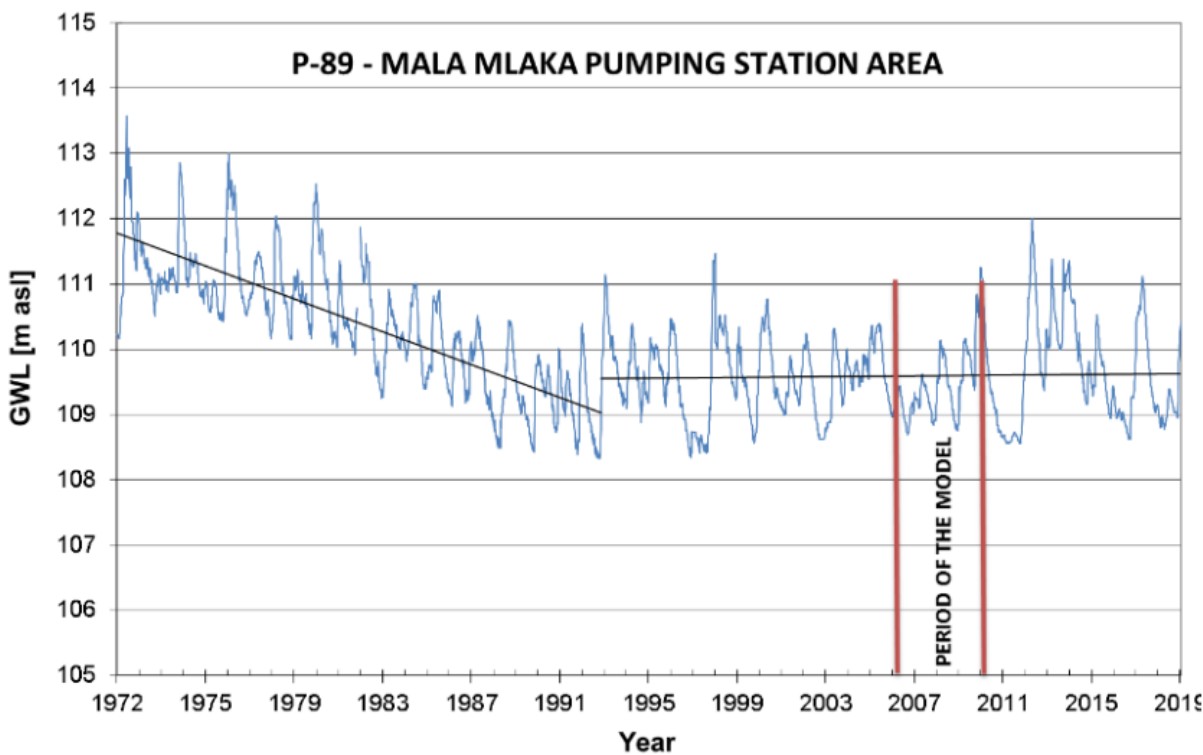

**Figure 5.** Groundwater levels in the Mala Mlaka pumping station area (piezometer P-89).

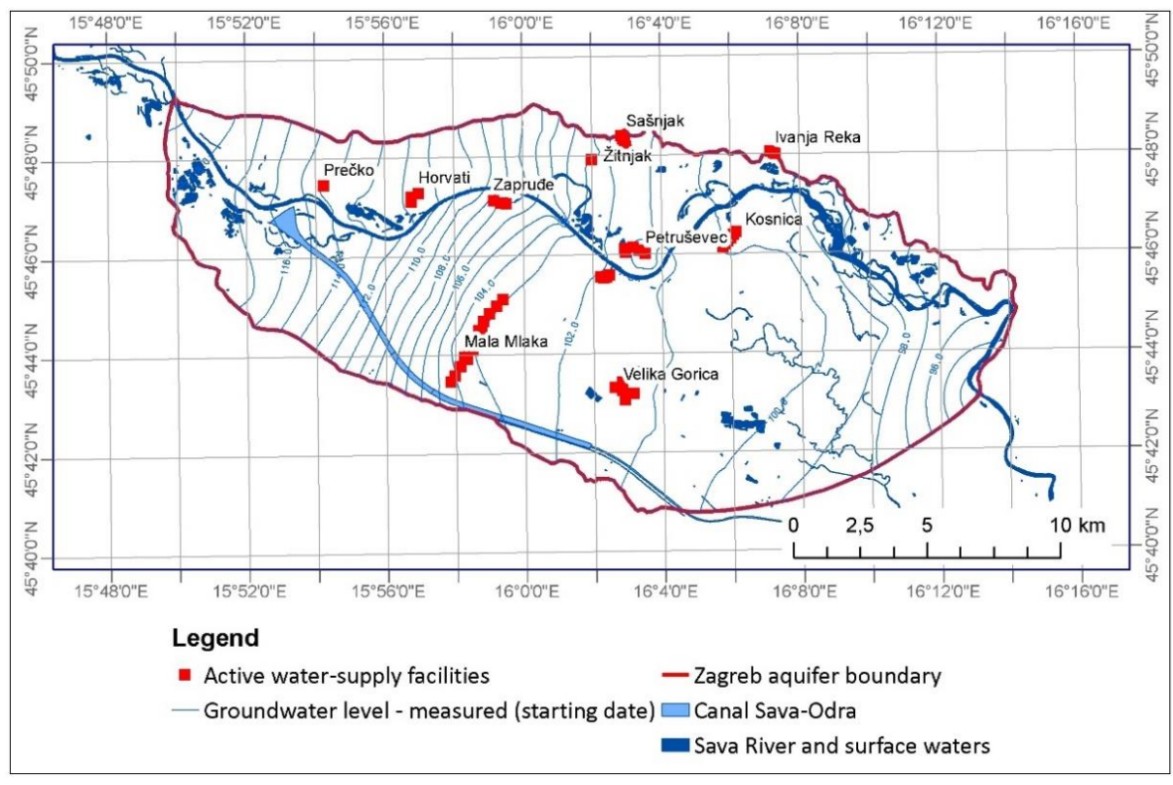

**Figure 6.** Initial distribution of groundwater levels for the starting date of the model.

*2.6. Model Calibration and Validation*

The calibration of a groundwater flow model is a procedure that examines the accuracy of a mathematical model. The modeled and actual groundwater levels at observation points in piezometric wells were compared.

It was necessary to review and process the existing database critically of all available piezometers and their measured water levels, and to remove from the calibration process, all those piezometers for which incomplete or questionable values of measured groundwater levels were observed. Usually, the periods for which the data were most complete and in which there were no extreme water phenomena (extremely low groundwater levels, extremely high groundwater levels, floods, etc.) were selected for calibration, but we wanted to include high and low water conditions, that is, at least one hydrological year.

For calibration purposes, 53 evenly distributed piezometers were selected throughout the Zagreb aquifer, and special attention was paid to the narrower area of the Mala Mlaka water pumping station, where piezometers located next to the wells were selected as calibration points (Figure 7).

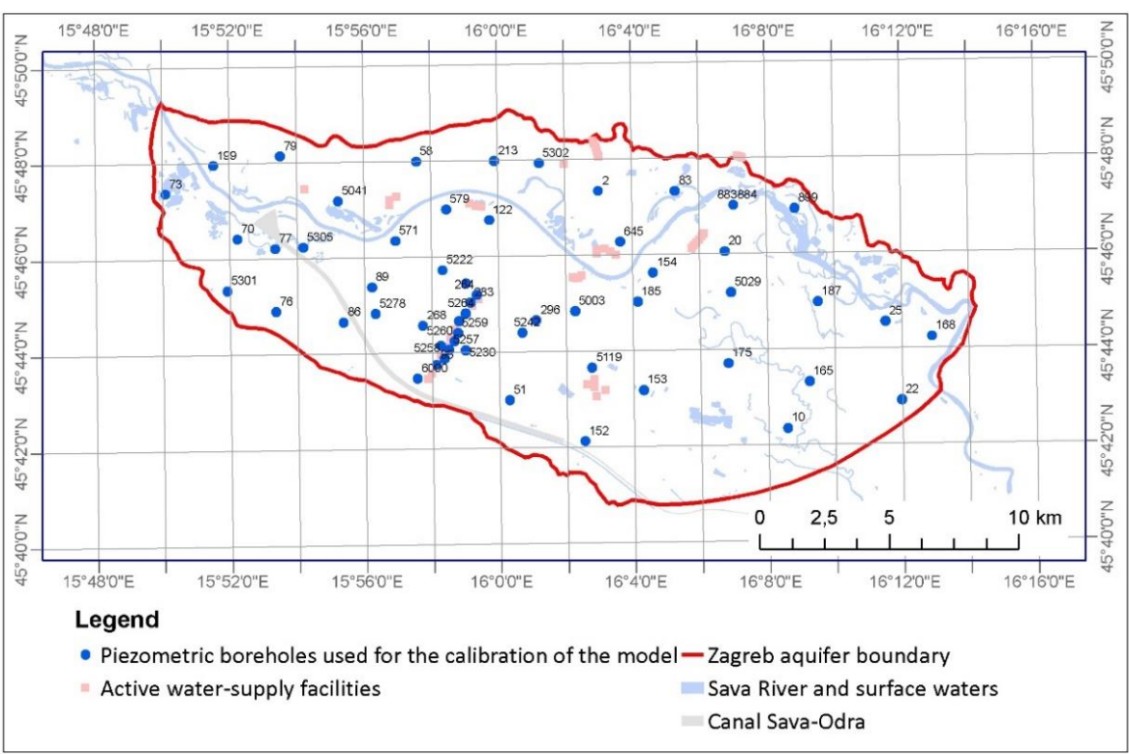

**Figure 7.** Spatial distribution of piezometric wells used for model calibration and validation.

In the process of model calibration, the biggest challenge was to achieve the best possible match between the model and the actual state, comparing the simulated (modeled) water level values at the control points with the actually measured water levels at these same points. This was largely successfully achieved by adjusting the model parameters during the calibration process: hydraulic conductivity, transfer rate parameter for the Sava River, marginal inflows, the influence of precipitation, and other factors. With each change in an individual parameter, the model was restarted until satisfactory results were achieved. The analysis showed satisfactory results with the coefficient of determination ($R^2$) value of 0.9937 (Figure 8a) and the root mean square error (RMSE) value of 0.3312.

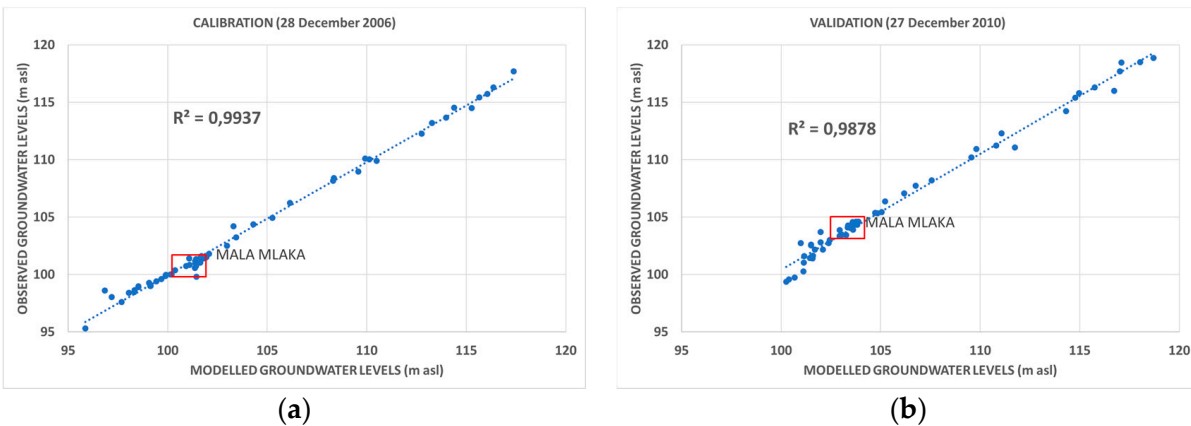

**Figure 8.** Calibration (**a**) and validation (**b**) of the model.

Following calibration, the model was validated using field data from 27 December 2010 on the same set of piezometers, and we compared them to the modeled data. The analysis showed satisfactory results with an $R^2$ value of 0.98 (Figure 8b).

### 3. Analysis of Groundwater Levels at the Mala Mlaka Water Pumping Station

The exploitation wells B-1 to B-10 of the Mala Mlaka water pumping station (Figure 9) were built during the 1960s and 1970s when the groundwater levels of the Zagreb aquifer were up to 5 m higher than today [11]. Then, the filter constructions were defined so that the upper elevations of the filter constructions were 1–3 m below the groundwater level in the dry period. However, the upper elevations of the filter structures nowadays remain in the unsaturated aquifer zone during dry periods as a result of the lowering of the groundwater levels in individual wells, significantly reducing the capacity of these wells. Therefore, at the end of the 1980s, the construction of additional drilled wells of greater depths (B-15, B-16, B-17, B-18, B-19, and B-21) began with the above-mentioned dug wells (Figure 9). From them, water is pumped into dug wells during low groundwater levels because there are pumps in the dug wells that are connected directly to the water-supply system.

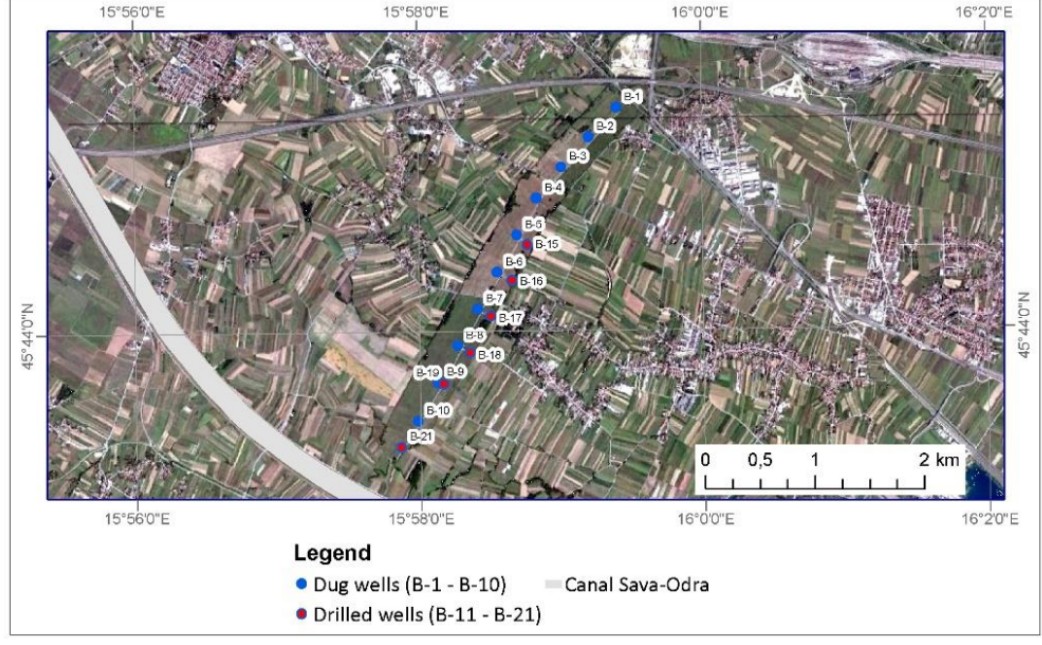

**Figure 9.** Wells of the Mala Mlaka water pumping station.

Based on the mathematical model of the Zagreb aquifer, each of the exploitation wells of the Mala Mlaka water pumping station was analyzed individually, i.e., the relationship between groundwater levels and the level of filter construction during the research period 2006–2010. For that period, measured and modeled groundwater levels at these wells were compared.

In the study period, during the dry periods when the groundwater levels were the lowest, several wells of the Mala Mlaka water pumping station experienced problems because their filter construction remained in the unsaturated zone of the aquifer. These are wells B-5, B-6, B-7, and B-8, which are located in the central part of the water pumping station (Figure 9). During the study period, the lowest groundwater level was recorded on 17 September 2007, whereas the maximum level was recorded on 30 December 2010 (Figure 10).

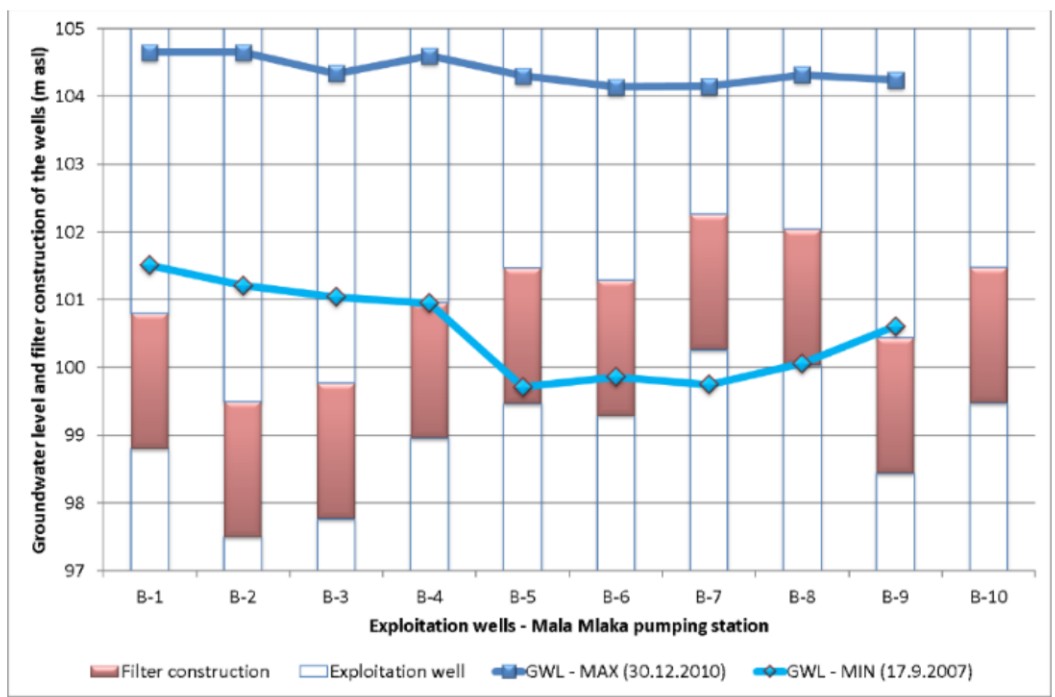

**Figure 10.** Minimum and maximum groundwater levels and filter construction of exploitation wells at the Mala Mlaka water pumping station.

The exploitation dug well B-5 of the Mala Mlaka water pumping station (Figure 9) is 14.70 m deep, and the filter structure is in the interval 12.70–14.70 m. In absolute elevations, the filter structure is in the interval 99.47–101.47 m a.s.l. (above mean sea level); during the period 2006–2010, groundwater levels were repeatedly lower than the upper elevations of the filter structure, thus reducing the capacity of the well. The lowest piezometric level was 99.71 m a.s.l. in the summer of 2007 (Figure 10) when almost the entire filter structure dried up.

The exploitation dug well B-6 (Figure 9) is 15.00 m deep, and the filter structure is in the interval 12.80–14.80 m. In absolute elevations, the filter structure is in the interval 99.29–101.29 m a.s.l.; during the period 2006–2010, groundwater levels were repeatedly lower than the upper elevations of the filter structure, thus reducing the capacity of the well. The lowest piezometric level was 99.68 m a.s.l. in the summer of 2007 (Figure 10).

The exploitation dug well B-7 (Figure 9) is 14.00 m deep, and the filter structure is in the interval 11.80–13.80 m. In absolute elevations, the filter structure is in the interval 100.26–102.26 m a.s.l. During the period 2006–2010, groundwater levels were generally lower than the upper elevations of the filter structure, thus reducing the capacity of the well. The lowest piezometric level was 99.75 m a.s.l. in the summer of 2007, and then the

groundwater level was lower than the lower level of the filter structure. The modeled groundwater levels showed a very good matching to the measured values (Figure 11).

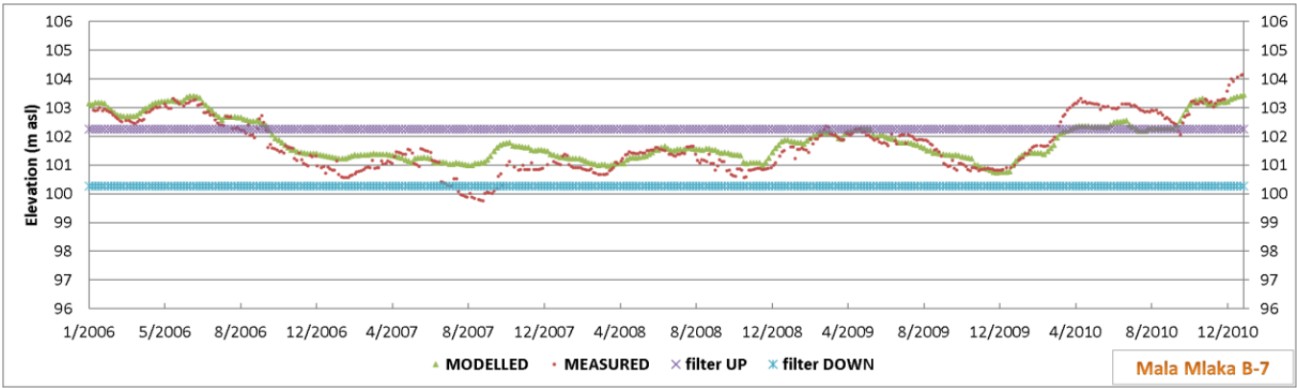

**Figure 11.** Relationship of measured and modeled groundwater levels with the position of the filter structure of exploitation dug well B-7.

The exploitation dug well B-8 (Figure 9) of the Mala Mlaka water pumping station is 14.00 m deep, and the filter structure is in the interval 11.30–13.30 m. In absolute elevations, the filter structure is in the interval 100.04–102.04 m a.s.l., and during the period 2006–2010, groundwater levels were generally lower than the upper elevations of the filter structure thus reducing the capacity of the well. The lowest piezometric level was 100.01 m a.s.l. in the summer of 2007, and then the entire filter structure was in the unsaturated aquifer zone (Figure 10).

The exploitation dug well B-10 (Figure 9) is 14.00 m deep, and the filter structure is in the interval 11.90–13.90 m. There are no piezometer boreholes next to well B-10 with which to compare groundwater level fluctuations with filter structure elevations, but modeled values indicate that during the period 2006–2010, groundwater levels were occasionally lower than the upper filter structure elevation (Figure 12).

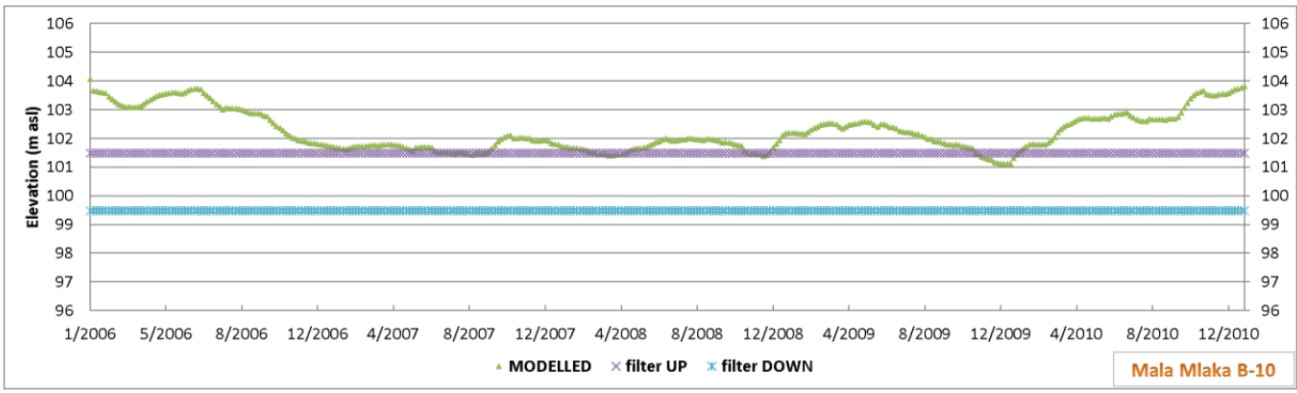

**Figure 12.** Relationship of modeled groundwater levels with the position of the filter structure of exploitation dug well B-10.

## 4. Results and Analysis of Various MAR Solutions

Based on the mathematical model of the Zagreb aquifer, six different variants of MAR were simulated; one was related with constant potential in the Sava-Odra Canal, three with the recharge from the Sava-Odra Canal with different backwater levels in the infiltration facility, and two with the construction of injection wells upstream of the Mala Mlaka water pumping station. Overall, the variant solutions can be divided into two basic groups: (a) MAR through the bed of the Sava-Odra Canal and (b) construction of injection wells upstream from the Mala Mlaka water pumping station.

This paper presents the effects of these MAR variants on the exploitation dug well B-7 of the Mala Mlaka water pumping station, where the negative impact of low groundwater levels of the Zagreb aquifer on the normal functioning of public water-supply was most pronounced.

(a)    MAR through the Bed of the Sava-Odra Canal

The Sava-Odra Canal was built after the extreme floods of the Sava River near Zagreb in 1964 to protect the city of Zagreb from floods. It was built so that it bypasses the city on the south side and manages the 1000-year high waters of the Sava River of 1510 m³/s. The Sava-Odra Canal is rarely used, only when the water levels of the Sava River exceed the level of the dam at the entrance to the Sava-Odra Canal. The projected flow of the Sava River at the time of the canal activation was 1900 m³/s, but due to the erosion of the Sava riverbed, the current flow of the Sava at the time of the canal activation was 2350 m³/s [36]. Since its construction in 1970 until today, the canal has been in operation only a few times: twice in 1979, in 1980, 1990, and 1998; in the period of 2006–2010, it was used only once in the fall of 2010. To increase the functionality of the canal, it was proposed to lower the dam at the entrance to the canal by ensuring that the canal begins to be active at a flow of the Sava River of 1900 m³/s [36].

Several MAR variants in the area of the Mala Mlaka water pumping station were simulated with the mathematical model of the Zagreb aquifer. One of the options is MAR from the Sava-Odra Canal while maintaining a constant potential of 0.5 m, but this requires larger construction works on the overflow structure and the entire length of the canal, as well as transferring part of the Sava River water to the canal.

Due to the need for large amounts of water in the Sava River, the option of building a dam with a height of about 2 m in the Sava–Odra Canal at the height of the Mala Mlaka water pumping station of 113 m a.s.l. was considered (Figure 13). This dam would ensure the constant presence of water in the canal in the narrower area of the Mala Mlaka water pumping station. The MAR was simulated for water levels in the canal of 115, 114.5, and 114 m a.s.l. which enable a water level in the canal of 1–2 m in the immediate zone of the water pumping station.

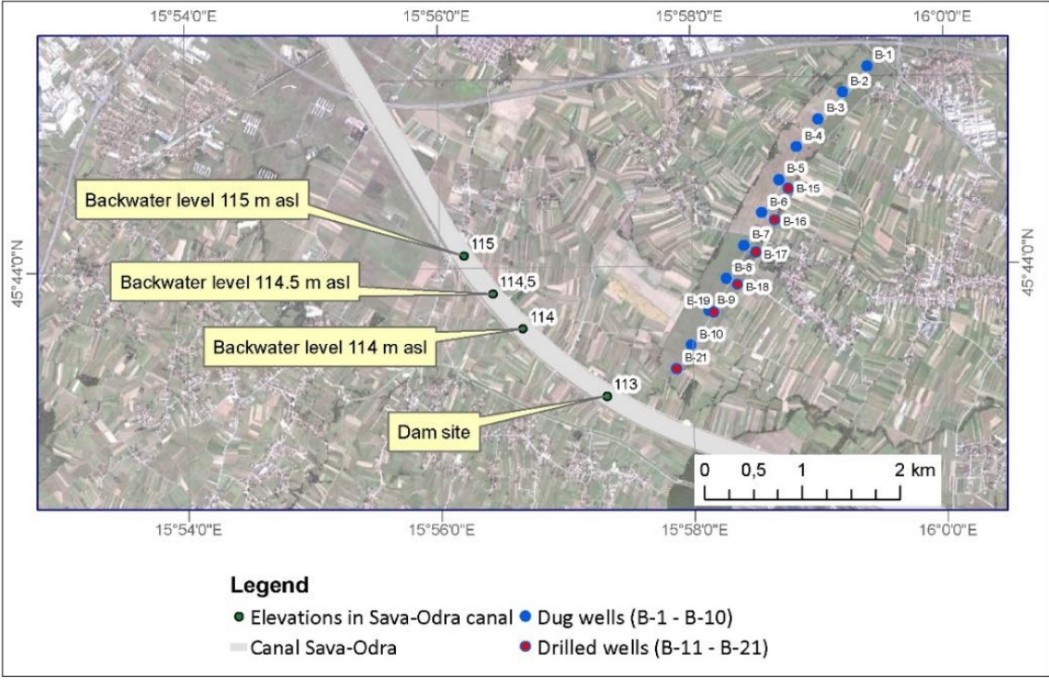

**Figure 13.** Position of MAR variants through the riverbed of the Sava-Odra Canal in the zone of the Mala Mlaka water pumping station.

A water level of less than 1 m in the canal was not considered due to the questionability of the implementation of such recharge. Water levels higher than 115 m a.s.l. were not realistic due to the limitation of the water level of the Sava River at the entrance to the Odra canal during the summer months when the demand for recharge is largest [37].

In the area of the Sava-Odra Canal in the considered section around the Mala Mlaka water pumping station, the bottom of the canal consists of cover deposits of the Zagreb aquifer. According to the data from the lithological database [2], the thickness of these deposits ranges from 1.5 to 3 m, which is important for defining the impact of infiltration through cover deposits and determining the value of the transfer rate parameter in the mathematical model. This part of the Sava-Odra Canal was simulated as the third type of boundary condition, with an assessment of infiltration into the aquifer (Cauchy boundary condition). In the mathematical model, the infiltration volume is calculated over the area, the transfer rate parameter, and the difference between the reference level and the actual groundwater level.

Estimates of the hydraulic conductivity of cover deposits, the clogging layer for the Sava-Odra Canal, are $8.2 \times 10^{-5}$ m/s [38], i.e., calculated at about 7 m/day. According to several studies [24,36], the vertical hydraulic conductivity of the Zagreb aquifer deposits is 10 times lower, which is about 0.7 m/day. The thickness of the cover deposits in the pumping zone is a maximum of about 3 m, which produces a transfer in parameter value of about 0.2 day$^{-1}$, which was used in the simulations.

The impact of the Sava-Odra Canal under the condition of a constant water level in the canal of 0.5 m did not have as strong an impact on the Mala Mlaka water pumping station as on the wider area, i.e., the relatively wide zone around the canal. The impact on well B-7 of the Mala Mlaka water pumping station is an increase in groundwater levels by about 1.8 m. This well experiences many problems during low groundwater levels because the groundwater level falls below the upper edge of the filter construction, and during the extreme dry periods, the whole filter construction remains dry. Applying the MAR of the Sava-Odra Canal with a constant water level of 0.5 m, the levels would be about 0.5 m above the upper edge of the filter (Figure 14).

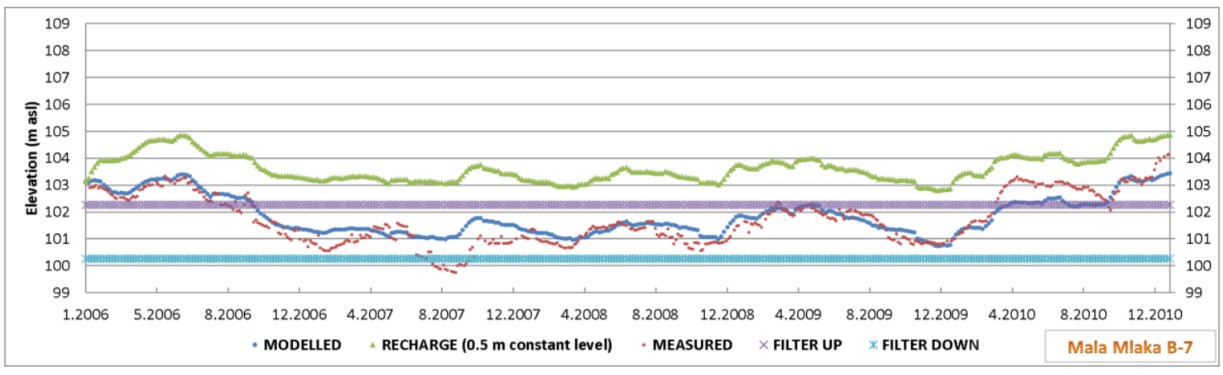

**Figure 14.** Influence of the MAR variant with a constant level of 0.5 m in the Sava-Odra Canal on well B-7 of the Mala Mlaka water pumping station.

By achieving a backwater level in the infiltration facility of 115 m a.s.l., the surface of the flooded terrain would be about 350,200 m$^2$ (2060 m long × 170 m wide), i.e., with an average height in the infiltration basin of 1 m the volume of the accumulation would be about 350,200 m$^3$. The rough estimate of the amount of water required for infiltration while maintaining a constant elevation in the reservoir of 115 m a.s.l. at a vertical infiltration of 0.7 m/day is about 2.8 m$^3$/s, to which evaporation should be added.

The influence of the MAR facility with a constant water level at an elevation of 115 m a.s.l. is important for the wider area of the Mala Mlaka water pumping station. The impact on well B-7 of the Mala Mlaka water pumping station is an increase in groundwater levels by about 4 m. Applying the MAR with a water level in the Sava-Odra Canal of

115 m a.s.l. the groundwater levels would be about 3.2 m above the upper edge of the filter (Figure 15).

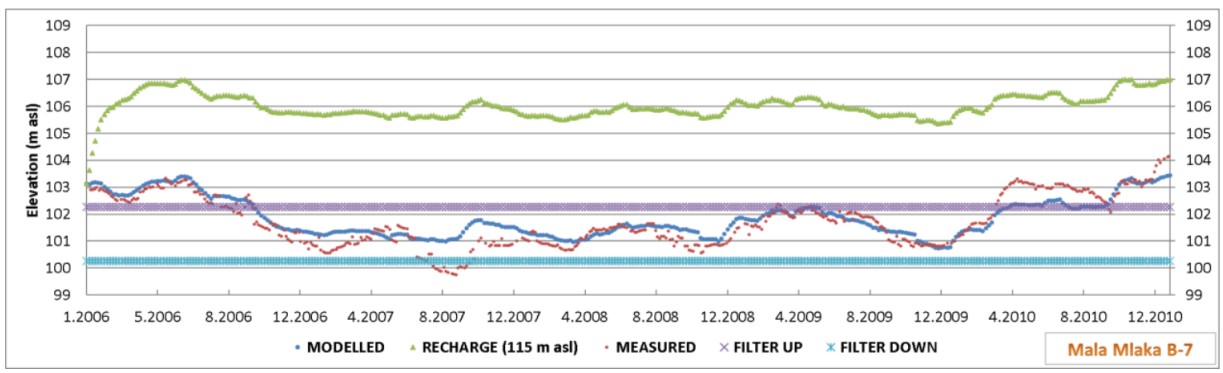

**Figure 15.** Influence of the MAR variant with a water level of 115 m a.s.l. on well B-7 of the Mala Mlaka water pumping station.

By achieving a backwater level in the infiltration facility of 114.5 m a.s.l., the area of the flooded terrain would be about 280,500 m$^2$ (1650 m long × 170 m wide), i.e., with an average height in the infiltration basin of 0.75 m the volume of the accumulation would be about 210,375 m$^3$. The rough estimate of the amount of water required for infiltration while maintaining a constant elevation in the reservoir of 114.5 m a.s.l. at a vertical infiltration of 0.7 m/day is about 2.3 m$^3$/s, to which evaporation should be added.

The influence of an artificial infiltration facility with a constant water level at an elevation of 114.5 m a.s.l. is slightly less than that of the MAR variant with an elevation of 115 m a.s.l. but is still significant for the wider area of the Mala Mlaka water pumping station. The impact on well B-7 of the Mala Mlaka water pumping station is an increase in groundwater levels by about 3.6 m. By artificial recharge with an elevation of 114.5 m a.s.l., the groundwater levels would be about 2.8 m above the upper edge of the filter (Figure 16).

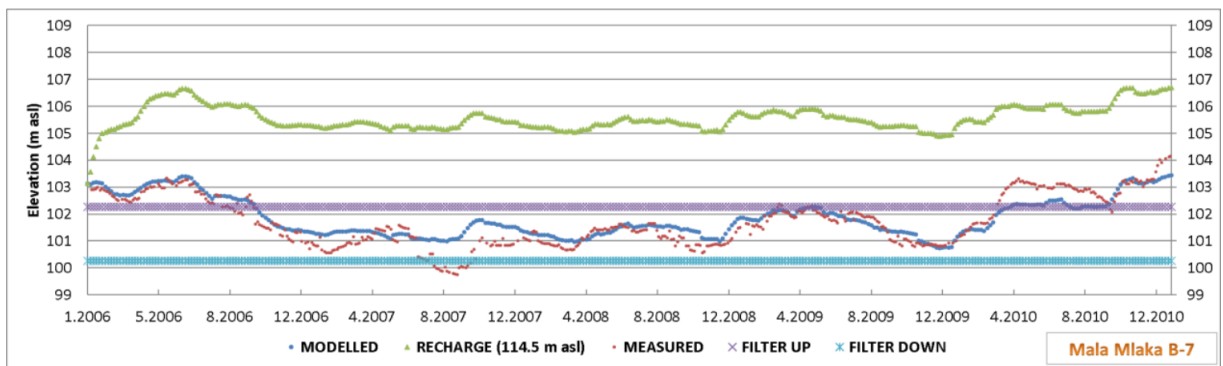

**Figure 16.** Influence of the MAR variant with a water level of 114.5 m a.s.l. on well B-7 of the Mala Mlaka water pumping station.

By achieving a backwater level in the infiltration facility of 114 m a.s.l., the area of the flooded terrain would be about 187,000 m$^2$ (1100 m long × 170 m wide), i.e., with an average height in the infiltration basin of 0.5 m, the volume of the accumulation would be about 93,500 m$^3$. A rough estimate of the amount of water required for infiltration with this elevation in the reservoir at a vertical infiltration of 0.7 m/day is about 1.5 m$^3$/s, to which evaporation should be added.

The influence of an artificial infiltration facility with a constant water level at an elevation of 114 m a.s.l. is the same on the wider area of the Mala Mlaka water pumping station. The impact on well B-7 of the Mala Mlaka water pumping station is the increase in groundwater levels by about 2.5 m. By artificial recharge with an elevation of 114 m a.s.l.,

the groundwater levels would be about 1.7 m above the upper edge of the filter during summer dry periods (Figure 17).

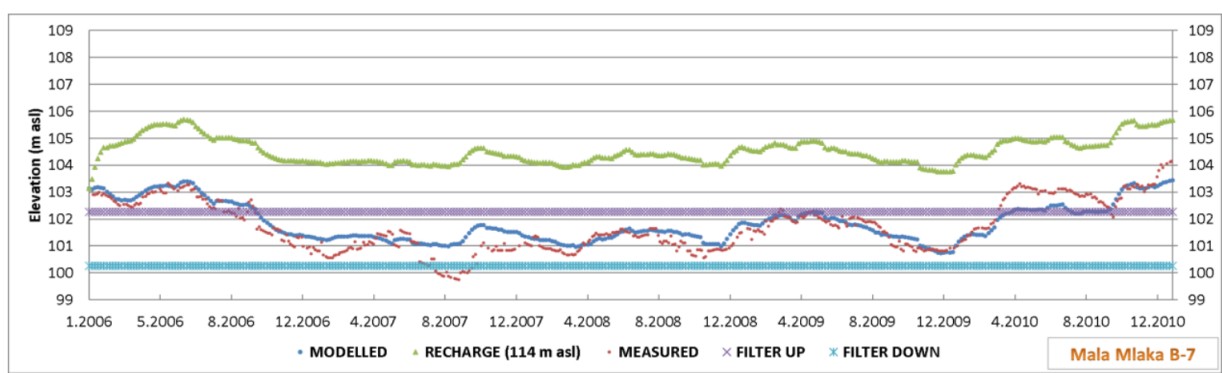

**Figure 17.** Influence of the MAR variant with a water level of 114 m a.s.l. on well B-7 of the Mala Mlaka water pumping station.

(b) Construction of Injection Wells

One of the considered options for artificial recharge is the construction of injection wells upstream of the exploitation wells of the Mala Mlaka water pumping station (Figure 18), which would ensure the self-purification of water before reaching the exploitation wells.

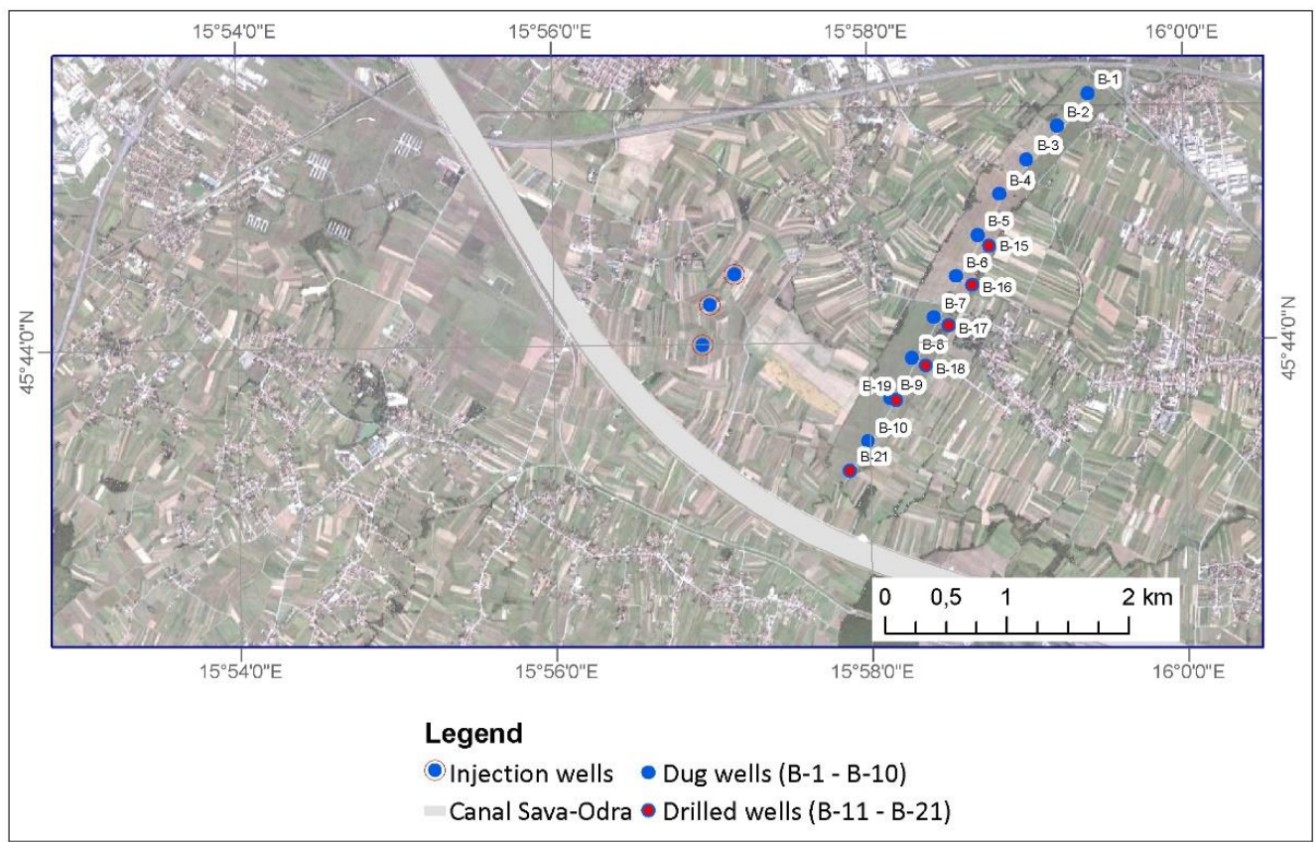

**Figure 18.** Position of the injection wells upstream from the Mala Mlaka water pumping station.

Two variants of artificial recharge using a battery of three injection wells with a capacity of 300 L/s and a capacity of 500 L/s were considered. The effect of injection

wells on the increase in groundwater levels at the Mala Mlaka water pumping station was examined by simulating the groundwater flow.

The impact of the three injection wells with a capacity of 300 L/s upstream from the Mala Mlaka water pumping station on well B-7 of the Mala Mlaka water pumping station is an increase in groundwater levels of about 1.3 m. Applying the MAR with three injection wells with a capacity of 300 L/s, the groundwater levels would be around the level of the upper edge of the filter structure during the summer dry periods, and with each over pumping, problems would occur due to the drying of the filter structure (Figure 19).

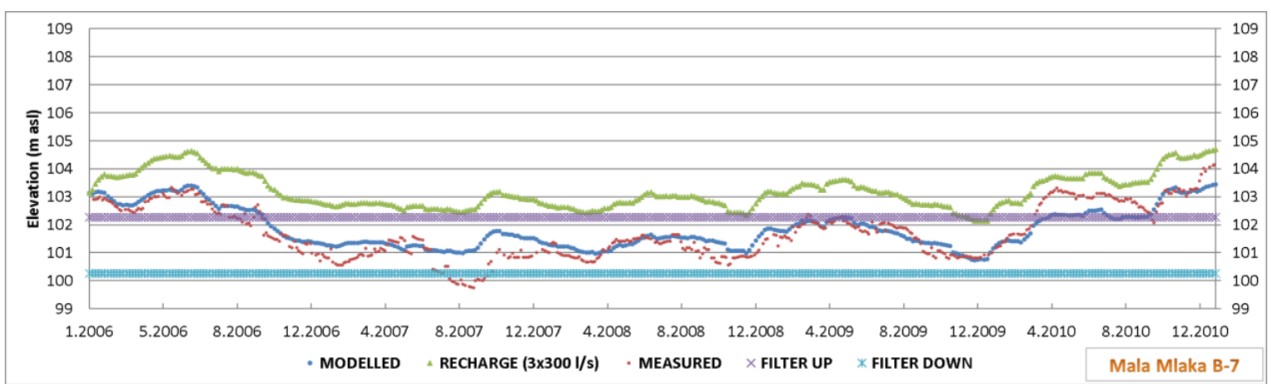

**Figure 19.** Influence of the MAR with injection wells (3 × 300 L/s) on the exploitation well B-7.

The impact of three injection wells with a capacity of 500 L/s upstream from the Mala Mlaka water pumping station on well B-7 of the Mala Mlaka water pumping station is an increase in groundwater levels of about 2 m. By artificial recharge with three injection wells with a capacity of 500 L/s each, the groundwater levels would be slightly less than 1 m above the level of the upper edge of the filter structure during summer dry periods (Figure 20).

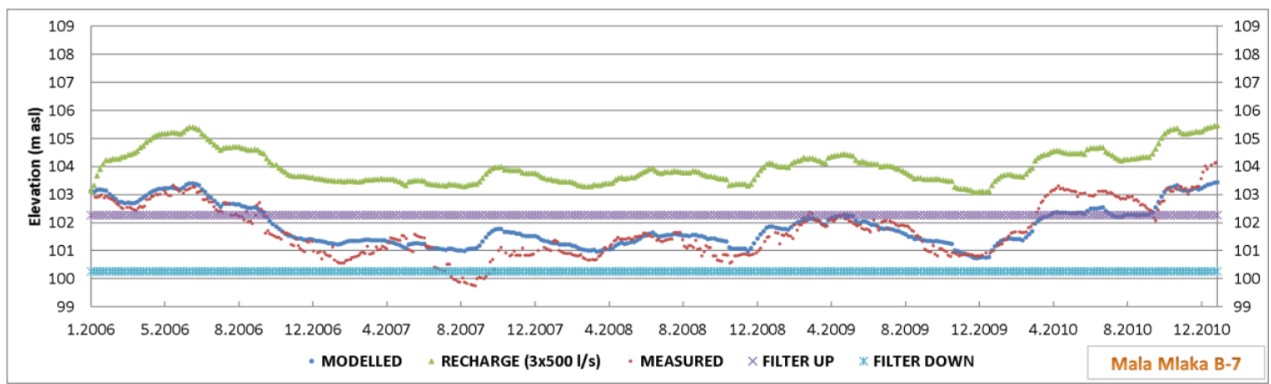

**Figure 20.** Influence of the MAR with injection wells (3 × 500 L/s) on well B-7.

## 5. Discussion and Conclusions

The Mala Mlaka water pumping station is located on the right bank of the Sava River, between the Sava-Odra Canal and the Sava River. The idea of building a water pumping station on this location started back in 1934, construction began in 1956, and the first wells started operation in 1964. Initially, 10 dug wells with a diameter of 6 m and a depth of 13 to 17 m were drilled, which provided 1.7 m$^3$/s of water for the water-supply system of the city of Zagreb.

At the time of the construction of the wells, the groundwater level of the Zagreb aquifer was a few meters higher than the upper elevations of the filters, which enabled the smooth operation of the water pumping station. From the 1980s until today, there has been a continuous decline in groundwater levels because of the erosion of the Sava riverbed and

the overexploitation of the aquifer. As a result, the groundwater level in some wells of the Mala Mlaka water pumping station is lower than the upper levels of the filter structures, which reduces the capacity of the water pumping station. This initiated the construction of additional and auxiliary drilled wells of greater depth, from which groundwater is pumped and transferred to the dug wells, and further pumped to the water-supply system. Eight auxiliary wells have been constructed with a depth of 31 to 41.3 m, with an elevation of the upper edge of the filter of 96.3 to 101.6 m a.s.l. Without the use of these auxiliary wells, the dug wells of the Mala Mlaka water pumping station would be out of order for most of the year today.

In the immediate vicinity, on the south side of the Mala Mlaka, a water pumping station passes the Sava-Odra Canal, which was built to protect Zagreb from flooding. The high waters of the Sava River enter the Sava-Odra Canal through the overflow structure and lower the water level of the Sava River in the area of Zagreb. From its construction until today, the Sava-Odra Canal has been in operation only a few times. The reason for this is the overflow level, which has changed due to the erosion of the Sava riverbed, so only extreme waters enter the canal. Therefore, the idea of a constant potential (constant water level) in the canal is not a realistic option without major construction modifications of the overflow facility and the entire length of the canal. The amount of water that would be necessary to maintain a constant water level in the canal would significantly affect the water levels of the Sava River during dry periods. Nevertheless, the initial simulation was conducted under the assumption of maintaining a constant potential (constant water level) in the Sava-Odra Canal. This simulation showed satisfactory results regarding the minimum groundwater levels in relation to the upper elevations of the filter structures of the wells. However, it requires extremely large amounts of water that can only be transferred directly from the Sava River. This would create water shortages in the Sava River during the summer dry periods and probably negatively affect the rest of the Zagreb aquifer.

Therefore, additional simulations were performed with the assumption of the construction of a 2 m tall barrier facility in the Sava-Odra Canal at the Mala Mlaka water pumping station at an elevation of 113 m a.s.l. This would allow a maximum water level in the artificial recharge facility of 115 m a.s.l., or 2 m at the barrier site. Since the Sava–Odra Canal has a fall, upstream from the barrier point, the depth of water in the infiltration facility would decrease. This would ensure the constant presence of water in the canal in the narrower area of the Mala Mlaka water pumping station. The water of watercourses, which passes through the canal on the south side of the Sava-Odra Canal and flows into the canal, can also be used to fill the infiltration facility.

To simulate the artificial recharge of the Zagreb aquifer from the Sava-Odra Canal, a mathematical model of the Zagreb aquifer was constructed and calibrated according to the measured values of groundwater levels from selected piezometric boreholes distributed throughout the Zagreb aquifer. In the area of the Mala Mlaka water pumping station, piezometers located next to the wells of the water pumping station were also used for calibration additionally ensure accuracy. Data such as pumping rates, hydrogeological parameters of the Zagreb aquifer, groundwater levels, water levels of the Sava River, and local watercourses, as well as the geometry of the aquifers, were used as input parameters for the mathematical model. All of them were organized into a GIS project that facilitated their search and browsing. The model was calibrated for 2006 because the data for that year were the most complete. After the completion of the calibration process, the simulation was conducted for a period of 5 years (2006–2010), which served as the basis for various models of artificial recharge of the Zagreb aquifer.

Groundwater flow for the period of 2006 to 2010 was simulated for six different variants of artificial recharge (MAR). One assumes a constant potential in the Sava-Odra Canal, three are related to recharge from the Sava-Odra Canal with different backwater levels in the infiltration facility (elevations of 114, 114.5, and 115 m a.s.l.), and two with three absorption wells upstream of the Mala Mlaka water pumping station (injection of 300 L/s each and 500 L/s each).

Only one of the simulations produced insufficient results for the entire simulation period: the artificial recharge of the aquifer with three injection wells with a capacity of 300 L/s each; the other recharge methods provide satisfactory results. This means that in all hydrological conditions through the period 2006–2010, the upper elevations of the filter construction of the wells of the Mala Mlaka water pumping station remain below the groundwater level.

The most favorable method to recharge the Zagreb aquifer artificially is achieved with an infiltration facility using an elevation of 115 m a.s.l. The use of such a facility will enable the smooth operation of the water pumping station and the possibility of increasing the pumping quantities at the Mala Mlaka water pumping station for the future development of the area.

**Author Contributions:** Conceptualization, H.M. and R.B.; methodology, R.B. and H.M.; software, H.M., R.B., J.L., and D.O.; validation, H.M. and R.B.; formal analysis, H.M. and R.B.; investigation, R.B. and H.M.; resources, H.M. and R.B.; data curation, H.M. and R.B.; writing—original draft preparation, H.M. and R.B.; writing—review and editing, H.M., R.B., J.L., and D.O.; visualization, R.B. and J.L.; supervision, R.B.; project administration, R.B.; funding acquisition, R.B. and H.M. All authors have read and agreed to the published version of the manuscript.

**Funding:** This research was funded by CROATIAN WATERS, Grant Number 25-075/2011. The APC was funded by the Support for scientific research from the Ministry of Science and Education of the Republic of Croatia.

**Institutional Review Board Statement:** Not applicable.

**Informed Consent Statement:** Not applicable.

**Data Availability Statement:** Not applicable.

**Conflicts of Interest:** The authors declare no conflict of interest.

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
