# Peer review of "The Possibility of Managed Aquifer Recharge (MAR) for Normal Functioning of the Public Water-Supply of Zagreb, Croatia"

_water, doi:10.3390/w13111562_

Round 1

Reviewer 1 Report

This paper can be considered as an interesting scientific work dealing with the proposal of several variants of managed artificial recharge of the Zagreb aquifer, which are simulated to determine the most optimal solution for the normal functioning of one of the main water-supply sites of the City of Zagreb water supply system.

The subject is within the topics of the Water Journal. The manuscript is clearly written following a structure that contains analysis, results and discussion documented and presented in a quite informative and reliable way.

My recommendation is that this manuscript should be accepted in its present form.

Author Response

Many thanks to the reviewer for the positive review and we hope that the paper will be interesting to a wider hydrogeological audience.

Reviewer 2 Report

The hydrogeological presentation is poor and does no contain all needed or desired relevant data and the supporting knowledge to justify there is a multi-layer aquifer, even if it probably behaves as such. A large part of the manuscript deals with the groundwater flow model but is poorly presented, it seems not well calibrated, boundary conditions are not rigorously presented and there is no validation. The text contains avoidable repetitions and writting is in some cases confusing. What is presented relative to MAR is the discussion of two kinds of ideas to be developed in the future. There is no new contribution and only is presented what would be the effect along the data period, by means of figures that really are not needed and that can be substituted by the statistical synthesis of results. The captions of the figures are nor clear enough.

Author Response

Point 1: The hydrogeological presentation is poor and does no contain all needed or desired relevant data and the supporting knowledge to justify there is a multi-layer aquifer, even if it probably behaves as such.

Response 1: Considering the reviewer's recommendations, the introductory part of the paper as well as Chapter 2 have been supplemented with a more detailed hydrogeological presentation of the data (subchapter 2.1. Geological and hydrogeological characterization). In addition, individual parts of the aquifer complex of the Zagreb aquifer are described separately in more detail. The figures related to hydrogeological representations and conceptualization of the Zagreb aquifer have also been added or corrected in such a way that they now better define them.

We believe that the existence of a multi-layer aquifer is now better presented.

Point 2: A large part of the manuscript deals with the groundwater flow model but is poorly presented, it seems not well calibrated, boundary conditions are not rigorously presented and there is no validation.

Response 2: The conceptual model, as well as the description of the boundary conditions, is now additionally explained in more detail (subchapter 2.2. Conceptual Model and subchapter 2.3. Boundary Conditions). The figures related to boundary conditions have also been corrected in such a way that they now better define them. We hope that we have responded to the reviewer's comment that the boundary conditions are not rigorously presented.

Regarding the poorly presentation of the groundwater flow model and its calibration, we added more details related to the key parameters that were needed to establish the model, especially the parts related to boundary conditions, the impact of the Sava River, the impact of pumping wells.

Calibration, as well as validation of the model are now further explained with more detail within the subchapter 2.7. Model Calibration as well as with a new additional subchapter 2.8. Model Validation.

Since there were no more detailed remarks by the reviewer, regarding the mathematical model and its calibration, we hope that we have given an answer to the reviewer with these additions.

Point 3: The text contains avoidable repetitions and writting is in some cases confusing.

Response 3: We thank the reviewer for the observed repetitions in the paper. We accepted his remarks and made the necessary corrections, harmonizing the texts and rearranging them by adding individual paragraphs - mostly in the introductory part and in Chapter 2, where we noticed the most repetitions.

Point 4: What is presented relative to MAR is the discussion of two kinds of ideas to be developed in the future. There is no new contribution and only is presented what would be the effect along the data period, by means of figures that really are not needed and that can be substituted by the statistical synthesis of results.

Response 4: Regarding this reviewer's comment, we can only state that he is right because that was one of the purposes of this paper. We tried to give a scientific answer to the solution to practical problems concerning the water supply of the capital of the Republic of Croatia.

Given that we are not investors, our ideas and proposals can evolve into something new only if those who requested a solution to the problem really do something with it. So far, that has not been done, so the problem is still relevant.

Regarding the selected period for which we performed the simulation of possible MAR solutions, it was selected based on the hydrological conditions which are now better presented in the chapter 2.6.

Namely, with an additional hydrological analysis at the request of another reviewer, we additionally showed why this period was chosen and showed that this is actually a representative hydrological period.

Point 5: The captions of the figures are nor clear enough.

Response 5: This has been corrected in all figures.

Reviewer 3 Report

I appreciate your work and technical solutions. My suggestions:

  1. The data used for the model are collected 11 years ago. I encourage the authors to refer either to more recent data.
  2. I think it would be helpful for the readers if the authors rephrased : "The Sava river... confluence with the Danube river..." (row 38); "Data of the water level...is one of the ..." (row269);  "...so in some it is every day..." (row276); '...an additional 6 drilled wells were drilled..." (row 299).
  3. The text in the figures should be more clear (written with higher fonts).

Author Response

I appreciate your work and technical solutions. My suggestions:

Point 1: The data used for the model are collected 11 years ago. I encourage the authors to refer either to more recent data.

Response 1: With an additional hydrological analysis at the request of the reviewer, we additionally showed why this period was chosen and showed that this is actually a representative hydrological period. Measurements of groundwater levels in the Zagreb aquifer began in early 1972 and continue to this day. The series of groundwater levels on several piezometers in the area of the Mala Mlaka pumping station were analyzed, and in the period from 1972-2019, two characteristic periods with different trends were observed. The first was from the be-ginning of 1972 to 1994, when there was a continuous decrease in groundwater levels in the Zagreb aquifer, while after 1994 the groundwater level stabilized and no further de-cline was recorded. For the period of the mathematical model, a period of five years was chosen (2006-2010) in which there were no large water level variations, i.e. when groundwater levels were low but steady.

Point 2: I think it would be helpful for the readers if the authors rephrased : "The Sava river... confluence with the Danube river..." (row 38); "Data of the water level...is one of the ..." (row269);  "...so in some it is every day..." (row276); '...an additional 6 drilled wells were drilled..." (row 299).

Response 2: We thank the reviewer for noticing incorrectly worded sentences. The sentences were reformulated, after which the English language was proofread.

Point 3: The text in the figures should be more clear (written with higher fonts).

Response 3: This has been corrected in all figures.

Round 2

Reviewer 2 Report

MAR is perhaps not the appropriate designation but induced recharge with surface water management

With additions the authors have improved the manuscript and now the presentation and conditions are clearer.

It should be explained the groundwater level evolution. Choosing a period of quasi-steady levels help modelling but still leaves some doubts on the application to longer periods.

Author Response

Point 1: MAR is perhaps not the appropriate designation but induced recharge with surface water management.

Response 1:

We appreciate the reviewer's opinion and possible doubts about appropriate designation, however in the article are described several methods of artificial recharge of the Zagreb aquifer. One is by injection wells which is certainly one of the MAR variants. The second method includes three variants of construction of a barrier site in the Sava-Odra canal, which also belongs to MAR. The category of induced recharge with surface water management could include infiltration from the Sava-Odra canal by maintaining a constant water level.

In this paper, we decided to use MAR because the examples of artificial recharge of aquifers that we deal with largely correspond to the definition of MAR.

Point 2: It should be explained the groundwater level evolution. Choosing a period of quasi-steady levels help modelling but still leaves some doubts on the application to longer periods.

Response 2:

This is a very good observation by the reviewer, because we may not have explained in the clearest way why, in our opinion, the model is still relevant. Due to that, the groundwater level evolution is added and better explained in the article in the rows 417-433.